# Decision Challenges for Managing Acute Paediatric Infections: Implications for Antimicrobial Resistance

**DOI:** 10.3390/antibiotics12050828

**Published:** 2023-04-28

**Authors:** Eva M. Krockow, Sanjay Patel, Damian Roland

**Affiliations:** 1School of Psychology and Vison Sciences, University of Leicester, Leicester LE1 7RH, UK; 2Department of Paediatric Infectious Diseases and Immunology, Southampton Children’s Hospital, Southampton SO16 6YD, UK; sanjay.patel@uhs.nhs.uk; 3Paediatric Emergency Medicine Leicester Academic (PEMLA) Group, Children’s Emergency Department, Leicester Royal Infirmary, Leicester LE1 5WW, UK; dr98@leicester.ac.uk; 4SAPPHIRE Group, Department of Population Health Sciences, University of Leicester, Leicester LE1 7RH, UK

**Keywords:** antimicrobial resistance, antibiotics, antimicrobial stewardship, infection, paediatrics, cognitive bias, principal-agent theory

## Abstract

Overprescribing of antibiotics in paediatrics accounts for a significant proportion of inappropriate antibiotic use in human healthcare, thereby contributing to the global health emergency of antimicrobial resistance. Antimicrobial stewardship efforts are complicated by the unique social dynamics in paediatric healthcare, with a specific challenge being the prominent role of parents and carers who act as intermediaries between prescribers and paediatric patients. In this Perspective article concentrating on healthcare of the United Kingdom, we describe this complicated interplay of different decision stakeholders (patients, parents and prescribers), outline four dimensions of decision challenges (social, psychological, systemic and specific diagnostic and treatment challenges) and provide a number of theory-based strategies for supporting different stakeholders during the decision process, ultimately with the aim of improving antimicrobial stewardship. Key decision challenges for patients and carers include limited knowledge and experience of managing infections, which were exacerbated during the COVID-19 pandemic and frequently result in health anxiety and inappropriate health-seeking behaviours. Challenges for medical prescribers span societal pressures from prominent patient litigation cases, cognitive biases, and system pressures to specific diagnostic problems (e.g., age limitations of current clinical scoring systems). Strategies for mitigating decision challenges in paediatric infection management will need to include a range of context- and stakeholder-specific actions, including improvements of integrated care and public health education as well as better clinical decision tools and access to evidence-based guidelines.

## 1. Introduction

Antimicrobial resistance (AMR) is a global health threat that was associated with approximately 4.95 million deaths in 2019 alone [1]. Unless we drastically reduce our use of antibiotics, AMR is going to overtake cancer as a leading cause of death by 2050 [2]. Overprescribing of antibiotics in paediatrics accounts for a significant proportion of inappropriate antibiotic use in human healthcare [3,4,5]. Yet, behaviour change to reduce antibiotic overuse remains a challenge in paediatric practice.

Treatment decisions for suspected bacterial infections including sepsis are complicated by high levels of risk and uncertainty [6,7]. In paediatric care, a further challenge is added by specific social dynamics, which notably include the role of a parent/carer as intermediary between patient and doctor. This article aims to highlight key decision-making challenges that arise from the unique context of diagnosing and treating paediatric infections across all sectors of care. Understanding and addressing these challenges is paramount to optimising antibiotic use in paediatric care and tackling the global threat of AMR.

All doctor–patient relationships can be conceptualised as principal–agent relationships [8], where an agent (i.e., the doctor) provides a service or takes action on behalf of the principal (i.e., the patient). All principal–agent relationships share a number of inherent features that complicate decision making. These include a potential misalignment of interests between principals and agents, meaning that the agent may act against the principal’s wishes, and an asymmetry of information, which makes it difficult for a principal to evaluate the service rendered by the more competent agent. In the case of doctor–patient relationships, there is an additional issue of “double agency” [9]. This refers to the fact that doctors typically answer to both the patient and the hospital or a trust they work for. With more than one principal to satisfy, the potential for a conflict of interests increases. After all, the doctor works to satisfy patient wishes while adhering to strict guidelines of conduct set by their place of work. Finally, in the case of paediatric medicine, this principal–agent relationship is complicated even further through the existence of an intermediary (the parent/carer) who represents the principal in direct interactions with the agent and communicates their interests.

Notably, the roles of principal and intermediary are likely to be blurred at times, especially in cases of pre-teenage patients who lack the capacity to voice their own interests. In those cases, where the parent/carer takes on a particularly prominent role, it may be difficult to recognise the true principal in the relationship. We conceptualise the paediatric patient as the principal throughout, because core texts of ethical guidance including the United Kingdom’s (UK) General Medical Council’s (the medical regulator in the UK) “Good Medical Practice” [10] highlight the patient as key stakeholder and stipulate a doctor’s service to the patient (rather than any relevant carers) as their main responsibility. The existence of an intermediary adds another level of complexity to the paediatric doctor–patient relationship.

In order to understand the unique challenges to diagnosis of paediatric infections and appropriate antibiotic prescribing decisions, it is essential to consider the role of each stakeholder within the principal–agent relationship and identify the interplay of different factors that may bias judgement or impede an optimal decision process.

## 2. Decision Challenges

Figure 1 offers an overview of social, psychological, systemic and specific diagnostic or treatment challenges that affect the decision making of principals, intermediaries and agents in the context of managing paediatric infections. It was conceptualised using an iterative process of reviewing the relevant literature and discussing professional experience of the clinical authors. The cited references were chosen following a purposive literature review, guided by the authors’ existing knowledge in the field of research. We appreciate that this is not systematic, but as a perspective piece of work, this paper had the objective to prompt debate in this field. Below follows a narrative discussion of specific factors shaping the judgement and choices of each key stakeholder implicated in the decision process.

### 2.1. The Paediatric Patient (Principal)

Parents, clinicians and society as a whole conceptualise children as an inherently vulnerable patient group in need of special protection [11]. Paediatric patients, especially young children, are limited in their cognitive ability to accurately assess and communicate their own disease symptoms and their severity. Furthermore, they are likely to be disproportionately affected by the unknown environment of doctors’ practices and hospitals, which could heighten their distress and inflate symptoms [12]. Social labels of vulnerability as well as a biased experience and communication of symptoms are likely to lead to a disproportionate risk of overdiagnosis and subsequent overprescribing in the paediatric patient group.

### 2.2. The Parent/Carer (Intermediary)

Social pressures around good parenting (e.g., from personal networks or day care providers) increase parents’ likelihood to engage in health-seeking behaviours such as the consultation of clinicians [11]. Furthermore, parents and carers face high levels of uncertainty and anxiety when experiencing acute illness in their children [13]. Research suggests that many parents struggle to make appropriate decisions about seeking medical care for their children [13]. Inaccuracy of parental judgements is evidenced by a mismatch between parents’ judgements and objective early warning scores (EWS); studies have determined that up to 10% of parents report their child as the most unwell they have ever seen them with the scoring system being at the lowest possible acuity level and the child being discharged safely [14,15]. Unfortunately, parental judgements appear to have been further impacted by the COVID-19 pandemic. Extended periods of lockdown and limited social contact meant that children contracted fewer infections and parents had less opportunity to develop their experience and judgement of disease symptoms [16]. Additionally, parental judgements could be affected by ongoing changes in Britain’s family structure. This involves an increase in smaller, “nuclear” family households and a loss of multigenerational support, with research suggesting that closer relationships with extended family increase overall family functioning [17]. In addition, the high number of alternative pathways to seeking healthcare support (e.g., telephone services and urgent care hubs), some of which vary across regions, makes it difficult to determine the most appropriate approach to seeking medical attention [18].

Finally, the persuasiveness of health-seeking behaviours and actions may not be related to the actual accuracy of the parents’ judgement or the objective severity of their children’s symptoms. Research on the candidacy or eligibility for care suggests that certain psychosocial characteristics (e.g., social class, education levels and English language abilities) determine the level of healthcare attention and resources allocated to each patient [19]. Similar to court trials, where acquittals appear related to articulate representation by legal experts, healthcare admissions may depend on the eloquence of patients and their representatives.

Taken together, pressures around good parenting, high levels of parental anxiety, lack of experience with infectious diseases, and limited knowledge of different pathways to seeking healthcare support might lead to increased health-seeking behaviours such as overconsultation by parents and carers. Depending on the parents’ abilities to negotiate their children’s eligibility for care, overconsultation may lead to overdiagnosis of bacterial infections and consequent overprescribing of antibiotics.

### 2.3. The Prescriber (Agent)

Prescriber judgement is affected by societal pressures stemming from a prominent history of patient litigation in paediatric care. High court cases punishing the so-called negligence of clinicians permeate the media. A recent example includes the case of paediatrician Bawa-Garba, who missed symptoms of kidney failure in a 6-year-old patient with Down syndrome and was subsequently found guilty of manslaughter by gross negligence [20]. Many public case discussions of physician negligence fail to account for system failures and contribute to an unhelpful culture of blame while potentially discouraging physician adherence to a duty of candour and engagement in reflective learning practices [21]. This, combined with preconceptions about children’s vulnerability, is likely to heighten risk aversion and induce disproportionate fear of underdiagnoses in prescribers, thereby increasing a tendency to overestimate the risk of infection and therefore lowering the threshold for antibiotic prescription.

A range of psychological factors further promotes these tendencies for overdiagnosis and overtreatment. Medical prescribers often operate under conditions of limited information and undue time pressures. This decision context necessitates quick decision making under uncertainty and is inherently prone to the influence of cognitive biases. For example, when judging a patients’ medical history conveyed by the parent/carer, prescribers are likely to be unduly influenced by parents’ emotional reactions and misguided judgements [22] described above. This may lead to disproportionate influences of confirmation bias (e.g., trying to confirm a preconceived parental diagnosis) or anchoring effects (e.g., using parental information such as subjective judgements on drowsiness and irritability or self-reported temperature measures as anchor points to guide the diagnostic process). Additionally, framing effects are likely to occur, when parents relay information and emphasise irrelevant aspects of their children’s medical history. For example, by framing symptoms in line with media narratives around the risk of preventable sepsis deaths, parents might evoke disproportionate levels of risk aversion in prescribers [23]. Finally, the existing literature suggests doctors’ mistaken perceptions of patient demand for antibiotic treatment [11], which may increase their propensity for prescribing as a “path of least resistance”.

Finally, specific challenges related to diagnoses and treatment of paediatric infections add to the complexity of prescriber decision making. Many junior prescribers are likely to have gained limited clinical experience in the area of general infection management during the pandemic, and they are dependent on either local senior clinical advice or national clinical guidelines, which sometimes contain conflicting directions. Additionally, non-specialist prescribers such as primary care and Emergency Department clinicians may perceive medical prescribing for children as more challenging. Reasons for this might include limited availability of clinical scoring systems for distinguishing viral and bacterial infections in children and the fact that the FeverPain score for tonsillitis is not validated for children aged below three years [24]. In addition, when determining appropriate medication dosage, prescribers have to choose between weight-banded and age-banded guidelines [25], which adds complexity to the decision process. Further challenges include higher rates of microbial colonisation in young children, which impacts the diagnostic accuracy of pathogen-focused point-of-care tests [26,27,28]. Finally, contrary to adult care, a key criterion for paediatric treatment choices and subsequent patient adherence may be the palatability of the drug in question [29].

## 3. Addressing Decision Challenges for Managing Infections in Paediatric Care

Improving infection management in paediatric care requires careful consideration of different areas of decision challenges and the stakeholders who are implicated. Table 1 presents an overview of strategies, which are mapped against the theoretical framework of principal–agent relationships discussed in the Introduction. A narrative review follows below.

### 3.1. Integrate and Educate

With the majority of patient-related decision challenges pertaining to the psychosocial dimension, key recommendations for improvement centre around the delivery of more integrated care include greater engagement with traditionally under-served patient groups. Additional actions may include improvement of public education, for example, by integrating health education within the core school curriculum [30]. All school-aged children (and future parents) need to understand the differences between viruses and bacteria and be aware of core self-care advice for infants and children, so that the natural trajectory of common diseases is understood. Education in this area could help to address a lack of experience and knowledge that fuel anxiety and associated health-seeking behaviours.

Boosting the decision capacity of parents and carers likely requires a similar approach. An evidence-based suite of resources across a range of languages and materials needs to be freely available in all regions to address inexperience in managing children’s infections. A successful example was set by the “Healthier Together” programme [31], a community initiative which provides online health advice for parents in specific UK regions. The recent marked uplift in health-seeking behaviours associated with the surge in Group A Streptococcus (December 2022) reinforces the importance of proactively offering trusted resources to parents to help build confidence in detecting signs of illness that may require specific treatment such as antibiotics. Additional activities need to address biased media reporting of adverse paediatric patient outcomes to reverse the culture of blame created by public discussions of healthcare failures. More public debate is required to discuss which patient outcomes are potentially preventable and which are not. Furthermore, the impact of overcrowding in hospitals needs to become a matter of public interest, rather than just an issue managed at a local level.

### 3.2. Decision Support

In the context of medical prescribers, strategies to improve the management of acute paediatric infections need to span all for four dimensions of decision challenges discussed in this article. Social challenges such as the pressures resulting from prominent cases of patient litigation need to be addressed through increased social support of prescribers, for example, involving better access to expert prescriber advice and senior review [6]. The UK’s medical professional regulator, the General Medical Council, has released guidance highlighting that the context of decision making (e.g., reduced staffing levels) is taken into account, but the guidance lacks specific examples, making it difficult to gauge its practical value.

With regard to individual psychological challenges, access to evidence-based guidelines and decision support tools on infection management is likely to be paramount. A recent review of antimicrobial stewardship programmes indicates that decision support tools such as electronic treatment algorithms may contribute to a significant reduction in antibiotic use (approximately 15%), while more traditional approaches of training appear to have comparatively little impact [32]. A specific example of paediatric decision support in the UK context includes the “Paediatric pathways” online resource [33], which was jointly created by the British Society for Antimicrobial Chemotherapy and the UK national paediatric groups. It offers clinical guidance on the management of common paediatric infections (e.g., cellulitis and tonsillitis). At present, these are simply available as online clinical pathways. These could be converted into decision support tools that are made available within currently used antibiotic prescribing guidelines such as Microguide^TM^. The use of these and other relevant decision tools should undergo research to evaluate whether access to specific decision tools improves clinical outcomes which, if validated, could then become clinical quality indicators. Carefully designed decision tools not only provide prescribers with more detailed information, but also have the additional potential to reduce intuitive or heuristic decision making and thereby decrease the harmful influence of cognitive biases. Indeed, previous trials of computerised decision support systems have shown such tools’ potential to improve clinical decision making, although adherence to the system’s recommendations and correct system use are likely to be prerequisites [34].

To tackle systemic decision challenges, system leaders must prospectively test the impact of national guidance on clinician behaviour to ensure their specificity and sensitivity in practice as well as minimise unintended consequences (such as work-arounds). Finally, strategies to support prescriber choices in the context of specific paediatric diagnostic and treatment challenges need to include further development and evaluation of age-appropriate scoring systems and point-of-care tests validated in children.

## 4. Conclusions

Decision making for infection management in paediatric care is complicated by an intrinsically difficult relationship between multiple stakeholders. Each stakeholder—principal/patient, intermediary/parent and agent/prescriber—faces particular challenges pertaining to social, psychological, systemic and specific diagnostic or treatment factors. Only a nuanced approach, which takes into account each stakeholder’s perspective as well as the nature of the challenge, is likely to result in an improvement of current practice. Key strategies may include the better delivery of integrated care and health education for paediatric patients and their parents/carers as well as a shift in the public debate around adverse patient outcomes. Strategies to support prescribers need to include improved organisational safety-netting, provision of decision tools, and access to evidence-based guidelines and diagnostic tools for paediatric prescribing.

## Figures and Tables

**Figure 1 antibiotics-12-00828-f001:**
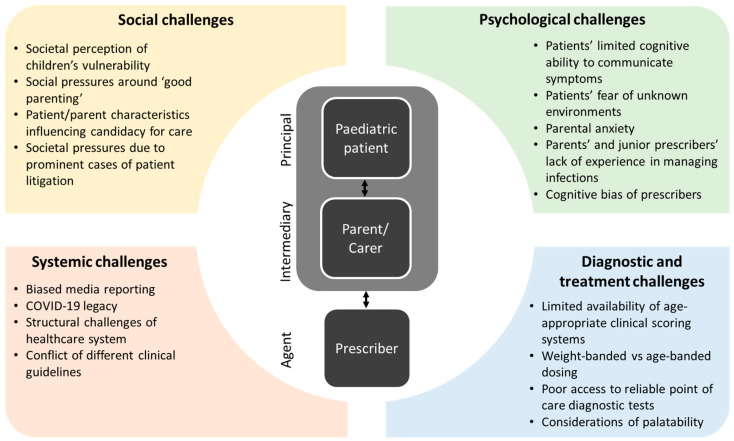
Different types of challenges affecting the judgement and choices of the key stakeholders implicated in the decision process around the management of paediatric infections.

**Table 1 antibiotics-12-00828-t001:** Recommendations for addressing key decision challenges across different stakeholders.

	Principal (Paediatric Patient)	Intermediary (Parent/Carer)	Agent (Prescriber)
Social challenges	Delivering integrated care through greater engagement with vulnerable and disadvantaged groups	Delivering integrated care through greater engagement with vulnerable and disadvantaged groups	Organisational safety-netting to minimise individual prescriber risks for litigation
Psychological challenges	Putting “health” on the core curriculum at school	Providing evidence-based suite of resources across a range of languages	Providing targeted decision tools (e.g., treatment algorithms)
Systemic challenges		Re-shaping public debates about litigation	Improved testing of national guidance to ensure specificity and sensitivity and minimise unintended consequences
Diagnostic and treatment challenges			Validating diagnostic tools and evidence-based guidelines for paediatrics

## Data Availability

There is no relevant data.

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
