# Peer review of "Decision Challenges for Managing Acute Paediatric Infections: Implications for Antimicrobial Resistance"

_antibiotics, 2023, doi:10.3390/antibiotics12050828_

Round 1

Reviewer 1 Report

Abstract

Based on the abstract, it seems that the logic issue may lie in the lack of clear connections between the different components of the framework presented. While the article aims to discuss the challenges of overprescribing antibiotics in paediatrics, the abstract jumps from discussing the global health emergency of antimicrobial resistance to the unique social dynamics of paediatric healthcare to the role of parents and carers as intermediaries.

Additionally, while the abstract outlines four dimensions of decision challenges, it is not clear how these dimensions are related to one another or how they contribute to the problem of overprescribing antibiotics. Moreover, the abstract presents a number of theory-based recommendations for addressing these challenges without explaining how these strategies might work or how they are connected to the different decision stakeholders and challenges outlined.

Overall, the abstract lacks clear structure and cohesion, making it difficult to understand how the different components of the framework fit together and contribute to the overall argument.

Introduction

In the introduction chapter, the authors briefly touch upon the impact of the COVID-19 pandemic on paediatric infection management, noting that the pandemic has exacerbated existing challenges such as limited knowledge and experience of managing infections, health anxiety, and inappropriate health-seeking behaviors. However, the authors do not provide much detail on how the pandemic has affected the issues discussed in the article or how it has influenced decision-making in paediatric infection management.

The main focus of the introduction chapter is on the principal-agent relationship in doctor-patient interactions and the additional complexity added by the role of parents and carers as intermediaries in paediatric care. The chapter provides a conceptual framework for understanding the unique challenges that arise from this relationship and the potential biases and conflicts of interest that can impact decision-making in paediatric infection management.

Overall, while the introduction chapter provides some useful background information on the principal-agent relationship and the role of intermediaries in paediatric care, it could benefit from a more detailed discussion of how the COVID-19 pandemic has affected decision-making in this context. This would help to establish the relevance of the article's focus on decision-making challenges in paediatric infection management to the current global health crisis.

Decision challenge

Based on the given excerpt, one aspect that could be improved for this chapter is to provide more specific examples and data to support the claims made. While the chapter does reference some research studies, it could benefit from including more detailed and concrete examples to illustrate the points being made. For example, the chapter discusses the impact of societal pressures on parents' health-seeking behaviors, but it does not provide specific examples of these pressures or how they manifest in real life. Similarly, the chapter mentions the influence of cognitive biases on prescriber decision-making, but it could be strengthened by providing more specific examples of how these biases can play out in practice. Including more specific examples and data could help to make the arguments presented in the chapter more compelling and relatable to readers.

Challenge & decision giving

One potential problem with the content is that it is quite dense and may be difficult for some readers to follow. The writing could benefit from more structure and clearer signposting to help guide the reader through the different points being made. Additionally, there are several acronyms used without explanation (e.g. UK General Medical Council) that may be confusing to some readers. Providing a brief explanation or link to further information would be helpful. Finally, there is a lack of specific examples given to illustrate some of the points being made, which could make it difficult for readers to fully grasp the significance of the recommendations being offered.

Overall

The concept and relationship among these four factors are not clearly identified, and for my personal point of view, all these factors are quite parallel, which is not dig into depth, and due to its unclear relationship between the concept and the special group of COVID pat

Author Response

Abstract

Based on the abstract, it seems that the logic issue may lie in the lack of clear connections between the different components of the framework presented. While the article aims to discuss the challenges of overprescribing antibiotics in paediatrics, the abstract jumps from discussing the global health emergency of antimicrobial resistance to the unique social dynamics of paediatric healthcare to the role of parents and carers as intermediaries.

Additionally, while the abstract outlines four dimensions of decision challenges, it is not clear how these dimensions are related to one another or how they contribute to the problem of overprescribing antibiotics. Moreover, the abstract presents a number of theory-based recommendations for addressing these challenges without explaining how these strategies might work or how they are connected to the different decision stakeholders and challenges outlined.

Overall, the abstract lacks clear structure and cohesion, making it difficult to understand how the different components of the framework fit together and contribute to the overall argument.

Response: We acknowledge the concerns raised and have aimed to link the ideas raised in the paper together. However, we respectfully feel that the purpose of the paper is to describe how these strategies link together and the abstract is simply to outline the aims of this paper (noting this is editorial/commentary piece rather than a piece of primary research)

Introduction

In the introduction chapter, the authors briefly touch upon the impact of the COVID-19 pandemic on paediatric infection management, noting that the pandemic has exacerbated existing challenges such as limited knowledge and experience of managing infections, health anxiety, and inappropriate health-seeking behaviors. However, the authors do not provide much detail on how the pandemic has affected the issues discussed in the article or how it has influenced decision-making in paediatric infection management.

The main focus of the introduction chapter is on the principal-agent relationship in doctor-patient interactions and the additional complexity added by the role of parents and carers as intermediaries in paediatric care. The chapter provides a conceptual framework for understanding the unique challenges that arise from this relationship and the potential biases and conflicts of interest that can impact decision-making in paediatric infection management.

Overall, while the introduction chapter provides some useful background information on the principal-agent relationship and the role of intermediaries in paediatric care, it could benefit from a more detailed discussion of how the COVID-19 pandemic has affected decision-making in this context. This would help to establish the relevance of the article's focus on decision-making challenges in paediatric infection management to the current global health crisis.

Response:

Comments along the same line have been made by other reviewers; we have removed the COVID19 aspect from the title as this is probably not relevant to the majority of the arguments made (although does play a role in changing parent expectation)

Decision challenge

Based on the given excerpt, one aspect that could be improved for this chapter is to provide more specific examples and data to support the claims made. While the chapter does reference some research studies, it could benefit from including more detailed and concrete examples to illustrate the points being made. For example, the chapter discusses the impact of societal pressures on parents' health-seeking behaviors, but it does not provide specific examples of these pressures or how they manifest in real life. Similarly, the chapter mentions the influence of cognitive biases on prescriber decision-making, but it could be strengthened by providing more specific examples of how these biases can play out in practice. Including more specific examples and data could help to make the arguments presented in the chapter more compelling and relatable to readers.

Response:  We have followed the reviewer’s suggestion to provide more specific examples for the influences of different cognitive biases. The revised paragraph reads as follows:

“For example, when judging a patients’ medical history conveyed by the parent/carer, pre-scribers are likely to be unduly influenced by parents’ emotional reactions and misguided judgements[23] described above. This may lead to disproportionate influences of confirmation bias (e.g., trying to confirm a preconceived parental diagnosis) or anchoring effects (e.g., using parental information such as subjective judgements on drowsiness and irritability or self-reported temperature measures as anchor points to guide the diagnostic process). Additionally, framing effects are likely to occur, when parents relay information and emphasise irrelevant aspects of their children’s medical history. For example, by framing symptoms in line with media narratives around the risk of preventable sepsis deaths, parents might evoke disproportionate levels of risk aversion in prescribers[24]. Finally, existing literature suggests doctors’ mistaken perceptions of patient demand for antibiotic treatment[11], which may increase their propensity for prescribing as a “path of least resistance”.”

We have also added some more concrete details to some of the other examples given in this section.

Challenge & decision giving

One potential problem with the content is that it is quite dense and may be difficult for some readers to follow. The writing could benefit from more structure and clearer signposting to help guide the reader through the different points being made. Additionally, there are several acronyms used without explanation (e.g. UK General Medical Council) that may be confusing to some readers. Providing a brief explanation or link to further information would be helpful. Finally, there is a lack of specific examples given to illustrate some of the points being made, which could make it difficult for readers to fully grasp the significance of the recommendations being offered.

Response:

Section headings have been used to break up this section into integration and education and then to decision support. The acronyms have been more fully explained.

Overall

The concept and relationship among these four factors are not clearly identified, and for my personal point of view, all these factors are quite parallel, which is not dig into depth, and due to its unclear relationship between the concept and the special group of COVID pat

Response:

We hope actions to your comments and the comments made by the other reviewers will produce a more refined piece of work.

Reviewer 2 Report

The manuscript provides opinion on the psychological and behavioural factors implicated in the increase of the antibiotic resistance that the society confronts nowadays. It is not an inedited manuscript; however, it emphasizes on some directions that should be addressed in order to have a better antibiotic stewardship in acute infection of the paediatric population.

As “perspective” article it may be suitable for this “special issue, however I think that the link with the “post COVID era” in the title is not covered by the substance of the article and I would suggest not to be used. Decision challenges for managing acute paediatric infections in the post COVID era: Implications for antimicrobial resistance.

In Figure 1, the “Psychological challenges” ideas 2-3 seem to be more or less the same, while idea 4 is unclear to me: what did authors want to say by “prescribers lack of experience” ?

All the analysed data makes reference to the structure and the organization of the UK Health System and this information is not present in the introduction or title of the manuscript draft. In the same time there are abbreviations which are not explained at the first use in the text e.g. emergency department = ED clinicians) pg.4, ln 160

Author Response

The manuscript provides opinion on the psychological and behavioural factors implicated in the increase of the antibiotic resistance that the society confronts nowadays. It is not an inedited manuscript; however, it emphasizes on some directions that should be addressed in order to have a better antibiotic stewardship in acute infection of the paediatric population.

As “perspective” article it may be suitable for this “special issue, however I think that the link with the “post COVID era” in the title is not covered by the substance of the article and I would suggest not to be used. Decision challenges for managing acute paediatric infections in the post COVID era: Implications for antimicrobial resistance.

Response:

We have removed COVID from the title as we agree this has been overemphasised.

In Figure 1, the “Psychological challenges” ideas 2-3 seem to be more or less the same, while idea 4 is unclear to me: what did authors want to say by “prescribers lack of experience” ?

Response:

We have amended the figure to address these points. The second psychological challenge focuses on patients, while the third challenge has been re-worded to emphasise a focus on parents. We amended idea 4 to highlight that it is mostly junior doctors who lack relevant experience.

All the analysed data makes reference to the structure and the organization of the UK Health System and this information is not present in the introduction or title of the manuscript draft. In the same time there are abbreviations which are not explained at the first use in the text e.g. emergency department = ED clinicians) pg.4, ln 160

Response:

This has been amended.

Reviewer 3 Report

I truly enjoyed reading your paper and congratulate you for a conceptually simple and reasonable thesis. Without considering the complexity around the moment of deciding and prescribing, it will indeed not be possible to tackle the causes of AMR.

Below are some suggestions that I believe could help you improve the strength of your work.

While I completely agree that it would be impossible to address the article’s topic with a systematic review, and I praise for the pertinence and usefulness of the authors’ personal perspective, at the same time I feel the need to understand better how this work was done. What was the method and criteria for the cited bibliographic references? How was Figure 1 built?

It is not clear to me what the authors mean by “theory-based recommendations”. What is/are the theory(ies) that serve as basis for the recommendations presented?

Also, the content in section “3. Addressing decision challenges for managing infections in paediatric care” sounds as strategies, general routes of action and not specific recommendations.

The abstract, Figure 1 and the Conclusions section are fine and convey the message.

Line 72 – “The existence of an intermediary adds another level of complexity to the paediatric doctor-patient relationship.” – the first sentence of the least paragraph of the Introduction belongs to the end o the previous paragraph (about the intermediary).

Line 141 – “shifting the baseline level for diagnosis in the opposite direction” is not clear - baseline level of what? What are the extremes of direction (i.e. opposite to what?)

Lines 220-222 – “The use of these and other relevant decision tools could become a clinical quality indicator, and further research should evaluate whether access to specific decision tools improves clinical outcomes. ” – it should be the other way around – first demonstrate impact on outomes and only then include as quality indicator.

Lines 223-225 – “Carefully designed decision tools not only provide prescribers with more detailed information, they have the additional potential to reduce intuitive or heuristic decision making and thereby decrease the harmful influence of cognitive biases.” – this sentence needs at least a reference.

MINOR ISSUES:

In Figure 1, there are some typos: a) “palatability” is missing an “I”; b) “poorer access” should be “poor access” – otherwise be clear about the comparator (poorer than what?); c) The title “Figure 1” is repeated.

Please review the references for format inconsistencies: for example a) the symbol “&” in the journal’s name of Ref 6 and 14; b) all the journals’ names – some show abbreviated and some full extension name, BMJ appears in at least 3 different formats, c) the author’s name in Ref 10 (GMC is not Council, G. M.!), d) references 19 and 20 are the same, I think; etc

Line 159 – parenthesis opens once and closes twice

Author Response

I truly enjoyed reading your paper and congratulate you for a conceptually simple and reasonable thesis. Without considering the complexity around the moment of deciding and prescribing, it will indeed not be possible to tackle the causes of AMR.

Response: Many thanks for this positive evaluation of our work.

Below are some suggestions that I believe could help you improve the strength of your work.

While I completely agree that it would be impossible to address the article’s topic with a systematic review, and I praise for the pertinence and usefulness of the authors’ personal perspective, at the same time I feel the need to understand better how this work was done. What was the method and criteria for the cited bibliographic references? How was Figure 1 built?

Response: Figure 1 was built using an iterative process of reviewing relevant literature and discussing professional experience of the clinical authors. The cited references were chosen following a purposive literature review, guided by the authors’ existing knowledge in the field of research. We appreciate this is not systematic but as a perspective piece of work the objective was to prompt debate in this field of work.    

It is not clear to me what the authors mean by “theory-based recommendations”. What is/are the theory(ies) that serve as basis for the recommendations presented?

Response: To add a clarification, we have revised the relevant sentence in the first paragraph of section 3 to read: “Table 1 presents an overview of recommendations, which are mapped against the theoretical framework of principal-agent relationships discussed in the Introduction.”

Also, the content in section “3. Addressing decision challenges for managing infections in paediatric care” sounds as strategies, general routes of action and not specific recommendations.

Response: We have amended this terminology.

The abstract, Figure 1 and the Conclusions section are fine and convey the message.

Response: Thank you for this positive feedback.

Line 72 – “The existence of an intermediary adds another level of complexity to the paediatric doctor-patient relationship.” – the first sentence of the least paragraph of the Introduction belongs to the end o the previous paragraph (about the intermediary).

Response: We have followed the reviewer’s suggestion and moved the indicated sentence to the previous paragraph.

Line 141 – “shifting the baseline level for diagnosis in the opposite direction” is not clear - baseline level of what? What are the extremes of direction (i.e. opposite to what?)

Response: The wording has been amended to “increasing a tendency to overestimate the risk of infection and therefore lowering the threshold for antibiotic prescription

Lines 220-222 – “The use of these and other relevant decision tools could become a clinical quality indicator, and further research should evaluate whether access to specific decision tools improves clinical outcomes. ” – it should be the other way around – first demonstrate impact on outomes and only then include as quality indicator.

Response: This has been amended

Lines 223-225 – “Carefully designed decision tools not only provide prescribers with more detailed information, they have the additional potential to reduce intuitive or heuristic decision making and thereby decrease the harmful influence of cognitive biases.” – this sentence needs at least a reference.

Response: Quite right. We have provided additional information and added a relevant reference: “Carefully designed decision tools not only provide prescribers with more detailed information, they have the additional potential to reduce intuitive or heuristic decision making and thereby decrease the harmful influence of cognitive biases. Indeed, previous trials of computerized decision support systems have shown such tools’ potential to improve clinical decision making, although adherence to the system’s recommendations and correct system use are likely to be prerequisites[35].

Akhloufi, H., van der Sijs, H., Melles, D.C. et al. The development and implementation of a guideline-based clinical decision support system to improve empirical antibiotic prescribing. BMC Med Inform Decis Mak 22, 127 (2022). https://doi.org/10.1186/s12911-022-01860-3

MINOR ISSUES:

In Figure 1, there are some typos: a) “palatability” is missing an “I”; b) “poorer access” should be “poor access” – otherwise be clear about the comparator (poorer than what?); c) The title “Figure 1” is repeated.

Response: We thank the reviewer for highlighting these issues with the figure. We have revised the figure as suggested.

Please review the references for format inconsistencies: for example a) the symbol “&” in the journal’s name of Ref 6 and 14; b) all the journals’ names – some show abbreviated and some full extension name, BMJ appears in at least 3 different formats, c) the author’s name in Ref 10 (GMC is not Council, G. M.!), d) references 19 and 20 are the same, I think; etc

Response: We are grateful for this detailed feedback. Most of the formatting issues are owed to the referencing software used. We will carry out a careful check of all references once the main contents of the article have been approved. We hesitate to make manual changes to the references at an earlier stage (while additional citations are still being added), because the reference software will revert references back to their original (faulty) state every time a new citation is added.

Line 159 – parenthesis opens once and closes twice

Response: Many thanks for highlighting this typo, which we have now corrected.

Round 2

Reviewer 1 Report

Thanks for authors' improvement, most of the concerns have been addressed. And understand that the paper is submitted under "Perspective", and the length of the presentation is limited.

With the help of the diagrams and tables, the logic structure and framework of the analysis have also been increased. Hence, I would suggest the paper be accepted in current form.

Author Response

Many thanks for your kind words regarding our paper. 

Reviewer 3 Report

THE PAPER IMPROVED AFTER REVISION. THANK YOU.

MINOR ISSUES:

While I completely agree that it would be impossible to address the article’s topic with a systematic review, and I praise for the pertinence and usefulness of the authors’ personal perspective, at the same time I feel the need to understand better how this work was done. What was the method and criteria for the cited bibliographic references? How was Figure 1 built?

Response: Figure 1 was built using an iterative process of reviewing relevant literature and discussing professional experience of the clinical authors. The cited references were chosen following a purposive literature review, guided by the authors’ existing knowledge in the field of research. We appreciate this is not systematic but as a perspective piece of work the objective was to prompt debate in this field of work.   

REVIEWER: IN MY OPINION IT IS NOT ENOUGH TO ANSWER DIRECTLY TO ME, BUT I NEED TO SEE THIS IDEA MORE CLEARLY IN THE PAPER.

Line 72 – “The existence of an intermediary adds another level of complexity to the paediatric doctor-patient relationship.” – the first sentence of the least paragraph of the Introduction belongs to the end o the previous paragraph (about the intermediary).

Response: We have followed the reviewer’s suggestion and moved the indicated sentence to the previous paragraph.

REVIEWER: I FIND IT IN THE SAME PLACE...

Author Response

Many thanks for the opportunity to respond again.

Response: Figure 1 was built using an iterative process of reviewing relevant literature and discussing professional experience of the clinical authors. The cited references were chosen following a purposive literature review, guided by the authors’ existing knowledge in the field of research. We appreciate this is not systematic but as a perspective piece of work the objective was to prompt debate in this field of work.   

REVIEWER: IN MY OPINION IT IS NOT ENOUGH TO ANSWER DIRECTLY TO ME, BUT I NEED TO SEE THIS IDEA MORE CLEARLY IN THE PAPER.

>>We have addressed this by inserting a version of our response to you into the main body of the text. 

Response: We have followed the reviewer’s suggestion and moved the indicated sentence to the previous paragraph.

REVIEWER: I FIND IT IN THE SAME PLACE...

>>apologies for this oversight. This has now been addressed.